# Clinical relevance of low-density *Plasmodium falciparum* parasitemia in untreated febrile children: A cohort study

**Mary-Anne Hartley**[1,2]*, **Natalie Hofmann**[3], **Kristina Keitel**[3], **Frank Kagoro**[4], **Clara Antunes Moniz**[3], **Tarsis Mlaganile**[4], **Josephine Samaka**[4,5], **John Masimba**[4], **Zamzam Said**[4], **Hosiana Temba**[4], **Iveth Gonzalez**[6], **Ingrid Felger**[3], **Blaise Genton**[1,3], **Valérie D'Acremont**[1,3]

**1** Centre for Primary Care and Public Health, University of Lausanne, Lausanne Switzerland, **2** EPFL, Machine Learning and Optimization Laboratory, Lausanne, Switzerland, **3** Swiss Tropical and Public Health Institute, University of Basel, Basel, Switzerland, **4** Ifakara Health Institute, Dar es Salaam, United Republic of Tanzania, **5** Amana hospital, Dar es Salaam, United Republic of Tanzania, **6** Foundation for Innovative New Diagnostics (FIND), Geneva, Switzerland

* Mary-Anne.Hartley@epfl.ch

**Data Availability Statement:** The data set and data dictionary are available from https://github.com/epfl-iglobalhealth/Hartley2020-PLoSMedicine.

## Abstract

### Background

Low-density (LD) *Plasmodium* infections are missed by standard malaria rapid diagnostic tests (standard mRDT) when the blood antigen concentration is below the detection threshold. The clinical impact of these LD infections is unknown. This study investigates the clinical presentation and outcome of untreated febrile children with LD infections attending primary care facilities in a moderately endemic area of Tanzania.

### Methods/findings

This cohort study includes 2,801 febrile pediatric outpatients (median age 13.5 months [range 2–59], female:male ratio 0.8:1.0) recruited in Dar es Salaam, Tanzania between 01 December 2014 and 28 February 2016. Treatment decisions were guided by a clinical decision support algorithm run on a mobile app, which also collected clinical data. Only standard mRDT+ cases received antimalarials. Outcomes (clinical failure, secondary hospitalization, and death) were collected in follow-up visits or interviews on days 3, 7, and 28. After patient recruitment had ended, frozen blood from all 2,801 patients was tested for *Plasmodium falciparum* (*Pf*) by ultrasensitive–quantitative polymerase chain reaction (qPCR), standard mRDT, and "ultrasensitive" mRDT. As the latter did not improve sensitivity beyond standard mRDT, it is hereafter excluded. Clinical features and outcomes in LD patients (standard mRDT-/ultrasensitive-qPCR+, not given antimalarials) were compared with those with no detectable (ND) parasitemia (standard mRDT-/ultrasensitive-qPCR-) or high-density (HD) infections (standard mRDT+/ultrasensitive-qPCR+, antimalarial-treated).

*Pf* positivity rate was 7.1% (*n* = 199/2,801) and 9.8% (*n* = 274/2,801) by standard mRDT and ultrasensitive qPCR, respectively. Thus, 28.0% (n = 76/274) of ultrasensitive qPCR+ cases were not detected by standard mRDT and labeled "LD". LD patients were, on

**Funding:** Analyses were funded by a Bill & Melinda Gates Foundation grant awarded to VDA https://www.gatesfoundation.org/ (grant number: OPP1163434). Laboratory work was supported by the Swiss National Science Foundation http://www.snf.ch/en/Pages/default.aspx (grant numbers 310030_159580 and IZRJZ3_164182). Malaria RDTs were contributed free of charge by Alere/Standard Diagnostics (now Abbott) https://www.alere.com/en/home/products-services/brands/sd-bioline.html The funders and donors had no role in study design, data collection and analysis, decision to publish, or preparation of the manuscript.

**Competing interests:** The authors have declared that no competing interests exist.

**Abbreviations:** ALMANACH, ALgorithms for the MANagement of Acute CHildhood illnesses; CRP, C-reactive protein; e-POCT, electronic point of care test; HD, high-density; HRP-2, *Pf* Histidine-Rich Protein-2; ICMI, Integrated Management for Childhood Illnesses; LD, low-density; mRDT, malaria rapid diagnostic test; MUAC, mid-upper arm circumference; ND, no detectable; PCT, procalcitonin; Pf, Plasmodium falciparum; qPCR, quantitative polymerase chain reaction; RR, risk ratio.

average, 10.6 months younger than those with HD infections (95% CI 7.0–14.3 months, $p < 0.001$). Compared with ND, LD patients more frequently had the diagnosis of undifferentiated fever of presumed viral origin (risk ratio [RR] = 2.0, 95% CI 1.3–3.1, $p = 0.003$) and were more often suffering from severe malnutrition (RR = 3.2, 95% CI 1.1–7.5, $p = 0.03$). Despite not receiving antimalarials, outcomes for the LD group did not differ from ND regarding clinical failures (2.6% [$n = 2/76$] versus 4.0% [$n = 101/2,527$], RR = 0.7, 95% CI 0.2–3.5, $p = 0.7$) or secondary hospitalizations (2.6% [$n = 2/76$] versus 2.8% [$n = 72/2,527$], RR = 0.7, 95% CI 0.2–3.2, $p = 0.9$), and no deaths were reported in any *Pf*-positive groups. HD patients experienced more secondary hospitalizations (10.1% [$n = 20/198$], RR = 0.3, 95% CI 0.1–1.0, $p = 0.005$) than LD patients. All the patients in this cohort were febrile children; thus, the association between parasitemia and fever cannot be investigated, nor can the conclusions be extrapolated to neonates and adults.

## Conclusions

During a 28-day follow-up period, we did not find evidence of a difference in negative outcomes between febrile children with untreated LD *Pf* parasitemia and those without *Pf* parasitemia. These findings suggest LD parasitemia may either be a self-resolving fever or an incidental finding in children with other infections, including those of viral origin. These findings do not support a clinical benefit nor additional risk (e.g. because of missed bacterial infections) to using ultrasensitive malaria diagnostics at a primary care level.

## Author summary

### Why was this study done?

- Every infection has a symptomatic threshold, above which the magnitude of infection burden triggers fever and other clinical symptoms that distinguishes ill individuals from those who remain asymptomatic. In malaria, this is called the "pyrogenic (fever) threshold."

- The standard malaria rapid diagnostic tests (sd-mRDTs), which have been in use for decades, can detect parasite loads of up to 10-fold below this threshold.

- New ultrasensitive malaria tests are now able to detect genetic traces of *Plasmodium* parasites that go 1,000-fold further.

- However, the clinical relevance of these minute levels of parasitemia is unknown.

- Does it detect a previously unappreciated cause of clinical failure? Or does it risk reporting a clinically irrelevant signal that may distract the clinician from other causes of fever?

### What did the researchers do and find?

- We assessed the clinical consequences of untreated low-density (LD) parasitemia that is only detectable with the ultrasensitive malaria test.

- In a cohort of 2,801 pediatric outpatients in Tanzania, we observed that of all patients in whom parasites were detected using ultrasensitive tests, over a quarter were not detected by standard mRDTs.

- However, despite not receiving antimalarials, patients with LD parasitemia did not differ from those without any detectable parasites regarding clinical outcomes during a 28-day follow-up.

## What do these findings mean?

- These findings suggest LD parasitemia may either be a self-resolving fever or an incidental finding in children with other infections.

- These findings neither support a specific utility nor risk (other than the obvious wasted resources) to using ultrasensitive malaria diagnostics at a primary care level.

## Introduction

The presence of *Plasmodium* parasites in the blood of a febrile patient does not necessarily imply causality. Low-density (LD), asymptomatic carriage of *Plasmodium* parasites is often more common than clinical malaria itself and may circulate in as many as 80% of individuals in highly endemic areas [1]. Indeed, the average parasitemia at which malaria-specific fever develops (known as the pyrogenic threshold) is estimated to be well over 50 parasites per microliter of blood [2]. LD infections are often below the detection threshold of standard malaria rapid diagnostic tests (standard mRDTs) and microscopy (at ±50 parasites/μl) [3–5]. Although these undiagnosed infections have the potential to serve as a reservoir for transmission [1], their impact on short- and long-term health outcomes is unknown. The utility and safety of detecting and/or treating such potentially harmless infections in febrile patients must still be evaluated.

Molecular diagnostic tools such as ultrasensitive quantitative polymerase chain reaction (ultrasensitive qPCR) are able to detect genetic traces of *Plasmodium falciparum* (*Pf*) parasites corresponding to infection densities as low as 0.03 parasites/μl [6], an over 1,000-fold improvement from the average 100–200 parasites/μl limit of standard mRDTs [7]. However, PCR-based testing requires trained personnel, well-equipped laboratories and other resources that are not always available in endemic areas. In response to this problem, a new easy-to-use and affordable "ultrasensitive" mRDT was developed (Alere, Abbott Diagnostics) able to detect the *Pf* histidine-rich protein-2 (HRP-2) with 10-fold greater sensitivity to that of standard mRDTs [8]. Our group, however, has reported only marginal improvements on sensitivity using ultrasensitive qPCR as a gold standard (75% versus 73%) [6]. It should be appreciated, however, that as HRP-2 is an inert protein component of *Pf* parasites, detection methods using HRP-2 are thus limited as proxy measures of parasitemia, in which thresholds of detection may vary greatly. Indeed, WHO-certified standard mRDTs are able to detect infections at approximately 200 parasites/ul but may be more sensitive in practice. So far, the ultrasensitive mRDTs have been trialed in various community surveys to assess *Pf* prevalence, but the clinical impact of such sensitive diagnostic tools has not yet been evaluated on symptomatic patients at the health facility level (where most patients with malaria are diagnosed).

As many of the LD infections are also asymptomatic, these highly sensitive tests challenge Koch's postulates of what defines a "pathogen" and at what concentration should a detected parasite be considered "disease-causing," necessitating the implementation of preventive or therapeutic interventions. It thus becomes important to assess at which point this gain in sensitivity surpasses its clinical benefit, perhaps even becoming counterproductive: where the detection of incidental *Pf* parasitemia in a febrile child may distract clinicians from diagnosing other serious and treatable coinfections such as bacterial sepsis or meningitis. A recent WHO assessment of the potential use of ultrasensitive mRDTs concluded that more studies are needed to evaluate the potential risk of missed diagnosis and treatment of serious illness following the identification of LD *Pf* infections and the potential benefits of detection and treatment of LD infections [7]. Currently, the WHO guidelines on the Integrated Management for Childhood Illnesses (IMCI) only recommend antimalarial treatment in the presence of a positive standard mRDT result [9]. Thus far, this strategy has proven to be safe in clinical practice and has considerably improved case management and rational use of antimalarial drugs both in endemic environments [10, 11] and in travelers (in whom parasite densities are typically lower) [12]. Because of the successful experience with standard mRDTs and their wide margin of safety, more sensitive methods are not being promoted for routine use in febrile case management.

To our knowledge, this is the first study to assess the clinical benefit and potential risk of using ultrasensitive malaria diagnostics at a primary care level in febrile children.

These analyses aim to fill gaps in policy recommendations that were highlighted by the WHO technical committee regarding the use of highly sensitive point-of-care *Pf* malaria diagnostics. We report the prevalence and clinical presentation of LD parasitemia detectable only by ultrasensitive tools such as ultrasensitive qPCR and ultrasensitive mRDT and evaluate its impact on clinical outcomes in a cohort of febrile children in a setting of moderate malaria endemicity.

## Materials and methods

### Design

**Patient population and context.** This secondary cohort analysis investigates data collected from 3,192 children (aged 2–59 months) with acute febrile illness (axillary temperature ≥37.5˚C for ≤7 days) recruited at 9 primary care outpatient clinics in Dar es Salaam, Tanzania between December 2014 and February 2016. The data were originally collected as part of a randomized controlled, noninferiority study to compare the clinical outcome of children using clinical decision support algorithms for the management of acutely ill children (e-POCT and ALMANACH) [13]. During the trial, the blood samples were collected for a prospective subcohort study investigating the etiologies and clinical relevance of nonmalarial febrile disease. The protocol for this trial was previously published here:

https://doi.org/10.1371/journal.pmed.1002411.s010. The details of which molecular tests would be used to elucidate infectious etiology from the biological samples were decided after the trial had concluded. Repeated malaria testing on blood samples was required after local health authorities reported suboptimal specificity of a single lot of standard mRDTs used onsite [14]. mRDTs as well as more sensitive methods of qPCR and "ultrasensitive" mRDTs were used to verify results. This study reports the clinical relevance of discrepancies between these test results.

All patients were assessed and managed using the electronic clinical decision support algorithms e-POCT [13] and ALMANACH [15], which offer diagnostic and treatment guidance derived from the WHO IMCI guidelines [9]. These IMCI-derived algorithms are designed to

guide clinicians through a structured consultation and recommend appropriate treatment predicted from the systematically collected data, including integrated point-of-care tests. No significant differences were noted between e-POCT and ALMANACH trial arms regarding the distribution of the outcomes or subcohorts studied.

Regarding malaria, all children were tested by standard mRDT; positive cases were treated with an antimalarial treatment, whereas negative cases were not given antimalarials.

**Aims.** The present study was designed to compare clinical outcomes between patients with untreated LD *Pf* parasitemia to those with no detectable (ND) parasites or those treated for high-density (HD) infection. We (1) measure the prevalence and distribution of LD parasitemia, (2) describe the clinical presentation of LD infection, and (3) investigate the impact of LD infection on clinical outcomes up to day 28.

## Clinical data and data collection

Data were thus collected via systematic clinical questionnaires embedded in a mobile app and included (1) demographic information (age, sex, region, date of consultation, etc.); (2) relevant medical history (HIV status, antibiotic or antimalarial consumption in the past 7 days, history of chronic illness, etc.); and (3) presence, severity, and duration of symptoms (fever, pain, or specific complaints of the respiratory, gastrointestinal, neurological, or dermatological system).

During the consultation, clinical signs of a relevant focused examination were also recorded (vital and danger signs, temperature, respiratory rate, malnutrition assessment, etc.) along with laboratory tests recommended by the app (mRDT, C-reactive protein [CRP], hemoglobin, urinary dipstick, etc.). The app then used this entered information to generate diagnoses (pneumonia, upper respiratory tract infection, malaria, presumed viral illness, etc.).

Finally, follow-up was performed physically at days 3 and 7 (except for children entirely cured at day 3 who were followed by telephone at day 7) and telephonically at day 28, to perform a follow-up assessment and management, if needed, and collect clinical outcomes (clinical failure, secondary hospitalization, or death). A physical visit was organized if the telephonic questionnaire revealed evidence of clinical failure. No patients were lost to follow-up in the subgroup of children included in the present analysis. The mobile algorithms are described in detail in the original paper [13]).

## Laboratory procedures

**Pf detection.** In order to ensure comparability of malaria tests, frozen whole blood samples were retested under standard laboratory conditions in Switzerland for the presence of *P. falciparum* after patient recruitment had ended. This included the 2 rapid tests: standard mRDTs (Malaria Ag Pf, Abbott Diagnostics; reference 05FK50; lot 05CDB228A) and ultrasensitive mRDT (Alere Malaria Ag *Pf*, Abbott Diagnostics; reference 05FK140; lot 05LDB004A), as previously described [14]. These rapid tests both targeted the HRP-2 antigen, the presence of which was verified in a previous study [14]. The ultrasensitive qPCR approach (described previously [14]) targeted the conserved C-terminal region of the multiple-copy *var* gene family, which has a limit of detection of <0.1 parasites per μL of blood [6]. As the ultrasensitive mRDT did not improve sensitivity beyond standard mRDT, it is excluded from further analyses.

## Exposures and outcomes

**Exposures.** The malaria test results were used to divide the population into 3 subgroups: (1) HD *Pf* infection (detected by standard mRDT and confirmed by ultrasensitive qPCR), (2)

**Table 1. Definition of the 3 subcohorts based on *Pf* parasitemia level and treatment received.**

| | | Malaria test results | | Received antimalarials |
| --- | --- | --- | --- | --- |
| | | Standard mRDT | Ultrasensitive qPCR | |
| Positive control | HD *Pf* parasitemia | + | + | Yes |
| Negative control | ND *Pf* parasitemia | − | − | No |
| Cases of interest | LD *Pf* parasitemia | − | + | No |

HD, high-density; LD, low-density; mRDT, standard rapid diagnostic test; ND, no detectable; qPCR, quantitative polymerase chain reaction.

LD *Pf* infection (ultrasensitive-qPCR+ samples that are not detectable by standard mRDT), or (3) not detectable by all available methods (ND) (Table 1). The administration of antimalarial treatment was based solely on the result of the standard mRDT performed on site at presentation and was thus given to children in the HD group. Children with medium-density (MD) *Pf* infection (ultrasensitive qPCR+ and ultrasensitive mRDT+ but standard mRDT-) could not be analyzed separately because of their low number (*n* = 3). Results of ultrasensitive mRDTs are presented in the supplement (S1 Table).

**Outcomes.** As described previously, morbidity and mortality outcomes were collected in follow-up visits or interviews at 3-, 7-, and 28-days postconsultation. Primary outcomes were the proportion of clinical failures (development of severe symptoms, significant dehydration or clinical pneumonia on/after day 3, or persistent symptoms at day 7) and severe adverse events (secondary hospitalizations, conversion to HD infection, and deaths) by day 28. Associations with clinical variables (signs/symptoms, diagnoses, and laboratory results) were also assessed.

Some exposures and outcomes are classified with levels of severity, and their definitions are tabulated in S2 Table along with the definition of clinical failure.

## Statistical analyses

Analyses were performed in Stata15 (SE, StataCorp, https://www.stata.com/) and presented using Prism 8 (GraphPad, https://www.graphpad.com/scientific-software/prism/).

**Epidemiology of *Pf* infection.** HD, LD, and ND are described in terms of prevalence and distribution across key demographic variables (age, sex, season). Fitted fractional polynomial plots were used to visualize predicted probabilities of each parasite density group across age.

**Clinical presentation of *Pf* infection.** The presence of clinical signs/symptoms as well as laboratory measures and final diagnoses in the LD group were compared with ND and HD (Table 1). Relative prevalence and bivariate analysis are reported as percentages and risk ratios (RRs) with *p*-values and confidence intervals. Logistic regression and chi-squared analyses were used. Risk ratios were obtained according to Zhang and Yu [16].

**Clinical outcomes.** Clinical outcomes in patients with LD infections were compared with ND and HD groups as described previously. This study primarily focused on comparisons between LD and ND groups, both of whom were not treated with antimalarials: A difference in outcomes between these groups would hence indicate LD *Pf* parasitemia-attributable morbidity. Dose-dependency of outcomes and presentations was also investigated using the parasite density estimated by ultrasensitive-qPCR. Logistic regression and chi-squared analyses were used as described previously.

**Controlling for bias.** Age, seasonality, and malnutrition have known independent associations with both *Pf* infection and the outcomes of interest; these parameters were controlled for confounding where appropriate and indicated whenever reported.

**Missing values.**   All missing data in variables used for analyses are indicated as absolute values in Tables 3–5. In general, missing values were generated because of the resource-conserving and clinical logic of the algorithms in the mobile app (i.e., not performing a certain test if the patient did not fulfil the pretest risk assessment criterion/a that would indicate the necessity of the test). For instance, hemoglobin was only measured in the 51% of patients with a clinical suspicion of anemia ($n = 1,374/2,801$ missing). Similarly, sickle cell anemia was only tested in 37% of the cohort ($n = 1,773/2,801$ missing). HIV test results were missing from 8.5% of the cohort ($n = 237/2,801$ missing) and district was missing in 4% of the cohort ($n = 110/2,801$). Missing values were missing at random according to the 3 subgroups (LD, HD, ND) and were thus excluded from analyses.

We present the missingness testing for the measurement of hemoglobin as it was particularly affected with only 51% of the values collected. Missingness was nonsignificant between both HD versus LD (47% versus 40% respectively, $p = 0.2$) and ND versus LD (49% versus 40% respectively, $p = 0.1$).

## Ethical considerations

Written informed caregiver consent was obtained at inclusion, and a sample donation form was filled out for sample storage and use for further evaluation of diagnostic methods.

Ethics approval was obtained from the Ifakara Health Institutional Review Board (IHI/IRB/EXT/16-2015), the Tanzanian National Institute for Medical Research (NIMR/HQ/R.8a/Vol. IX/1789), and the Swiss Ethikkommission Nordwest-und Zentralschweiz (EKNZ-UBE-15/03).

This study is reported as per the Strengthening the Reporting of Observational Studies in Epidemiology (STROBE) guideline (S1 STROBE Checklist).

## Results

### Cohort selection and *Pf* parasitemia prevalence

**Population.**   Of 3,192 recruited patients, 3,004 had sufficient sample volume for diagnostic analyses, and 203 were excluded because of discrepancies between onsite standard mRDT results and those performed later on frozen blood (**Fig 1**). These discrepancies were likely due to the suboptimal specificity of a single lot of standard mRDTs used onsite and were excluded to eliminate cohort contamination (i.e., those with a negative standard mRDT should not be exposed to antimalarials). The performance issues on this lot of standard mRDTs was independently reported by the local health authorities and verified by HRP2 antigen concentration in an associated study [14]. Of the 203 faulty tests, 84% ($n = 170/203$) had false positive onsite standard mRDT results (i.e. receiving antimalarials despite having ND *Pf* parasitemia) and 16% ($n = 33/203$) had false negative results (i.e. not receiving antimalarials despite having HD malaria). Fortunately, all the children in the latter category, who were put at risk of untreated HD malaria by the faulty tests spontaneously cleared the infection without treatment by day 7. The e-POCT (electronic point-of-care test) clinical decision support tool reacted to severity signs in this group of false negative children, referring 15% of them ($n = 5/33$) directly to hospital or for a next day re-consultation, significantly more than were referred in the malaria-free false positive group (3%, $n = 5/170$, $p = 0.003$).

The proportion of participants receiving antibiotics was 20.7% ($n = 523/2,527$), 15.8% ($n = 12/76$), and 23.7% ($n = 47/198$) in children with ND, LD, and HD infections, respectively. No significant differences in antibiotic prescription were observed between ND versus LD groups (RR = 0.7, 95% CI 0.4–1.3, $p = 0.3$) or HD versus LD groups (RR = 0.7, 95% CI 0.4–1.1, $p = 0.2$) and could thus not explain the differences described hereafter.

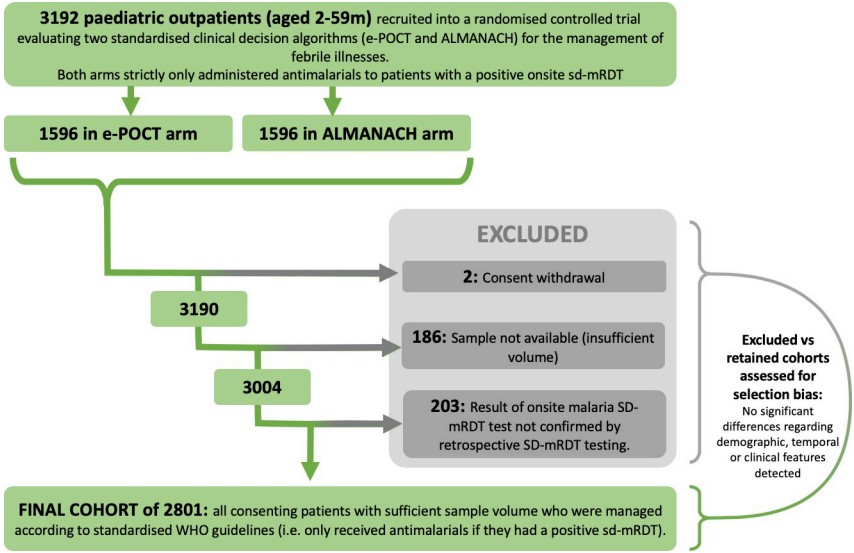

**Fig 1. Population flow chart.** After patient recruitment had ended, frozen blood from all 2,801 patients was tested for *Pf* by ultrasensitive qPCR, standard mRDT, and ultrasensitive mRDT. As the latter did not improve sensitivity beyond standard mRDT, it is hereafter excluded. ALMANACH, ALgorithms for the MANagement of Acute CHildhood illnesses; e-POCT, electronic point of care test; mRDT, malaria rapid diagnostic test; *Pf*, *Plasmodium falciparum*, qPCR, quantitative polymerase chain reaction; sd-mRDT,.

**Pf parasitemia prevalence.** Detailed performance measures of the 3 tests of interest (standard mRDT, ultrasensitive mRDT, and ultrasensitive qPCR) were previously described [14]. *Pf* positivity was 7.1% (*n* = 199/2,801) by standard mRDT, 7.5% (*n* = 209/2,801) by ultrasensitive mRDT, and 9.8% (*n* = 274/2,801) by ultrasensitive qPCR. Standard mRDT and ultrasensitive qPCR results are presented in **Table 2**; ultrasensitive mRDT results can be found in the supplement, **S1 Table**. The rate of "false positives" compared with ultrasensitive qPCR was 0.04% (*n* = 1/2801) by standard mRDT and 0.3% (n = 8/2801) by ultrasensitive mRDT. A total of 27.7% (*n* = 76/274) and 26.6% (*n* = 73/274) of ultrasensitive qPCR+ *Pf* infections were not detected by standard mRDT and ultrasensitive mRDT, respectively. Thus, the number of children with HD, LD, and ND was 198, 76, and 2,527, respectively.

## Distribution of *Pf* parasitemia in febrile children

**Demographic distribution.** LD infections disproportionally affected younger children, in whom the predicted probability of having LD infection was higher than that of HD infection

**Table 2. Results of standard mRDT and ultrasensitive qPCR tests.**

| | | us-qPCR (gold standard) | | Total |
|---|---|---|---|---|
| | | − | + | **Total** |
| sd-mRDT | − | 2,526 | <u>76</u> LD: us-qPCR-positive cases missed by sd-mRDT | 2,602 |
| | + | 1 | <u>198</u> HD: sd-mRDT-positive cases confirmed by us-qPCR | 199 |
| | | <u>2,527</u> ND: Total us-qPCR-negative samples | 274 | 2,801 |

HD, high-density *Pf* infection; LD, low-density *Pf* infection; ND, no detectable *Pf* parasitemia; sd-mRDT, standard malaria rapid diagnostic test; us-qPCR, ultrasensitive quantitative PCR.

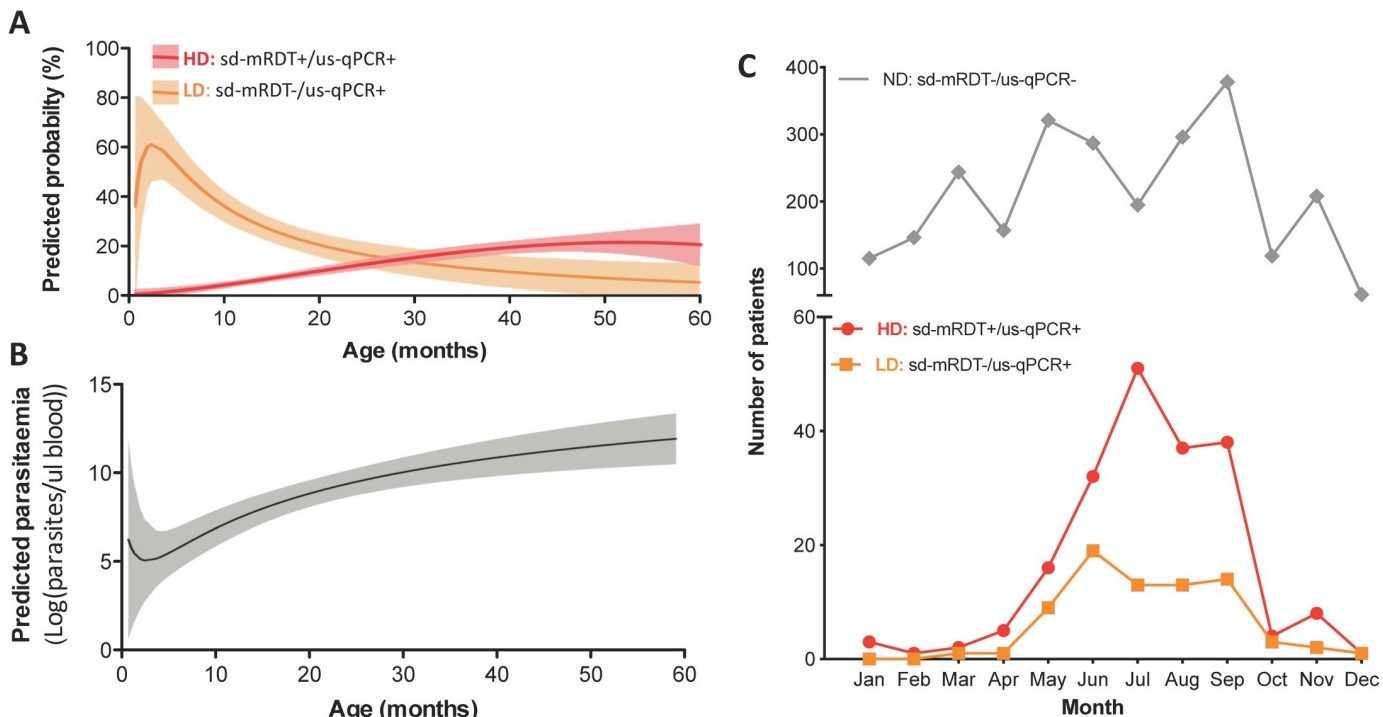

**Fig 2. Temporal and age distribution of patients with LD, HD, and ND parasitemia. (A)** Predicted probability of being assigned to the HD or LD group according to age (fitted fractional polynomial plot). Shaded regions are 95% CI; **(B)** Predicted probability of parasite density by age (fitted fractional polynomial plot); **(C)** Number of patients with HD and LD parasitemia identified for each month of the 2016 recruitment period. HD, high-density *Pf* infection, LD, low-density *Pf* infection; RDT, rapid diagnostic test; ND, no detectable *Pf* parasitemia; sd-mRDT, standard malaria RDT; us-qPCR, ultrasensitive quantitative polymerase chain reaction.

until 30 months of age (**Fig 2**). Patients with LD infections were on average 10.6 months younger than those with HD infections (95% CI 7.0–14.3 months, *p* < 0.001, median 13.1 versus 25.6 months), whereas no significant age difference was observed between LD and ND groups (95% CI 2.6–2.9 months, *p* < 0.9) (**Tables 3 and 4**). This correlation could also be seen at the parasite density level (**Fig 3**). No differences between male and females were observed among any groups (**Tables 3 and 4**).

**Seasonal distribution.** LD infections mirrored the seasonal distribution pattern of HD infections (**Fig 2**) occurring mostly during the post-rainy season with a peak between June and September. Compared with ND, LD carriage was 4.2-fold more likely to occur during the malaria season (95% CI 2.4–7.2, *p* < 0.001, **Table 3**). LD carriage dropped to near zero outside the malaria season.

Age was identified as a potential confounder in comparisons between HD and LD groups, whereas season was identified as a potential confounder in comparisons between ND and LD groups (**Tables 3–5**).

## Clinical presentation of children with *Pf* infections

**LD versus ND.** Overall, 43% (*n* = 33/76) of children with LD parasitemia were ascribed the diagnosis of undifferentiated fever (13% more than the ND group [29.8%, *n* = 753/2527], RR = 1.8, 95% CI 1.1–2.7, *p* = 0.01). In contrast, the LD group was less likely to have cough (RR = 0.5, 95% CI 0.3; 0.7, p = 0.001) and pneumonia (RR = 0.4, 95% CI 0.2; 0.8, *p* = 0.01) (**Table 5**) No differences amongst any groups were observed for gastrointestinal symptoms,

**Table 3. Comparative analysis of demographic and epidemiological features (ND versus LD).**

| | ND (Undetectable) sd-mRDT-/us-qPCR- | | LD parasitemia sd-mRDT-/us-qPCR+ | | LD versus ND parasitemia | | | | | | | |
| --- | --- | --- | --- | --- | --- | --- | --- | --- | --- | --- | --- | --- |
| | Prevalence | | | | Crude bivariate analysis | | | | Adjusted (Season) | | | |
| Total | % | Prevalence | % | Prevalence | | | | | | | | |
| n = 2,801 | 90.2 | 2527/2801 | 2.7 | 76/2801 | | | | | | | | |
| Demographics | Median | IQR | Median | IQR | RR$_{crude}$ | 95% CI | p-value | | RR$_{adj}$ | 95% CI | p-value | |
| Age (months) | 13.0 | (8.3–21.7) | 13.1 | (8.4–21.0) | 1.0 | (1.0–1.0) | 0.9 | | 1.0 | (1.0–1.0) | 0.7 | |
| | % | Total = 2,527 | % | Total = 76 | RR$_{crude}$ | 95% CI | p-value | | RR$_{adj}$ | 95% CI | p-value | |
| Sex (female) | 44.3 | 1,120 | 50.0 | 38 | 1.3 | (0.8–2.0) | 0.3 | | 1.3 | (0.8–2.0) | 0.3 | |
| Season (post rainy) | 50.5 | 1,275 | 81.6 | 62 | 4.2 | (2.4–7.2) | <0.001 | [a] | –> Potential confounder | | | |
| District (Kinondoni) | 60.2 | 1,457/2,422 | 57.7 | 41/71 | 0.9 | (0.6–1.4) | 0.7 | | 1.2 | (0.7–1.9) | 0.5 | |
| District (Temeke) | 32.1 | 778/2,422 | 42.3 | 30/71 | 1.5 | (1.0–2.4) | 0.07 | | 1.0 | (0.6–1.7) | 0.8 | |
| District (Ilala) | 7.7 | 187/2,422 | 0.0 | 0/71 | – | – | – | | – | – | – | |
| Treatment exposure | % | Total = 2,527 | % | Total = 76 | RR$_{crude}$ | 95% CI | p-value | | RR$_{adj}$ | 95% CI | p-value | |
| Antimalarials | 0.0 | 0 | 0.0 | 0 | – | – | – | | – | – | – | |
| Antibiotics | 20.7 | 523 | 15.8 | 12 | 0.7 | (0.4–1.3) | 0.3 | [a] | 0.8 | (0.5–1.5) | 0.6 | |

Low-density (LD: sd-mRDT-/us-qPCR+) versus undetectable *Pf* parasitemia (ND: sd-mRDT-/us-qPCR-).

HD, high-density *Pf* infection; LD, low-density *Pf* infection; RR, risk ratio; RR$_{adj}$; adjusted RR; RR$_{crude}$, crude (unadjusted) RR; sd-mRDT, standard malaria rapid diagnostic test; us-qPCR, ultrasensitive quantitative PCR.

[a]$p < 0.05$ and considered as statistically significant.

skin problems, pharyngitis, or clinical danger signs. **(Tables 5–6).** No difference was found between LD and ND groups regarding anemia (RR = 1.0, 95% CI 0.6; 1.9, *p* = 0.8) **(Fig 3)** or levels of inflammatory markers (CRP and PCT levels) **(Fig 4).**

**Table 4. Comparative analysis of demographic and epidemiological features (LD versus HD).**

| | HD parasitemia sd-mRDT+/us-qPCR+ | | LD parasitemia sd-mRDT-/us-qPCR+ | | LD versus HD parasitemia | | | | | | | |
| --- | --- | --- | --- | --- | --- | --- | --- | --- | --- | --- | --- | --- |
| | Prevalence | | | | Crude bivariate analysis | | | | Adjusted (Age) | | | |
| Total | % | Prevalence | % | Prevalence | | | | | | | | |
| n = 2,801 | 7.0 | 198/2,801 | 2.7 | 76/2,801 | | | | | | | | |
| Demographics | Median | IQR | Median | IQR | RR$_{crude}$ | 95% CI | p-value | | | | | |
| Age (months) | 25.6 | (15.3–37.4) | 13.1 | (8.4–21.0) | 0.9 | (0.9; 1.0) | <0.001 | [a] | → Potential confounder | | | |
| | % | Total = 198 | % | Total = 76 | RR$_{crude}$ | 95% CI | p-value | | RR$_{adj}$ | 95% CI | p-value | |
| Sex (female) | 48.0 | 95 | 50.0 | 38 | 1.1 | (0.6–1.8) | 0.8 | | 1.2 | (0.7–2.2) | 0.5 | |
| Season (post rainy) | 81.8 | 162 | 81.6 | 62 | 1.0 | (0.6–1.5) | 1.0 | | 0.9 | (0.5–1.5) | 0.7 | |
| District (Kinondoni) | 39.9 | 73/183 | 57.7 | 41/71 | 1.7 | (1.1–2.3) | 0.01 | [a] | 1.7 | (1.1–2.3) | 0.02 | [a] |
| District (Temeke) | 56.3 | 103/183 | 42.3 | 30/71 | 0.7 | (0.4–1.0) | 0.05 | [a] | 0.7 | (0.4–1.0) | 0.06 | |
| District (Ilala) | 3.8 | 7/183 | 0.0 | 0/71 | – | – | – | | – | – | – | |
| Treatment exposure | % | Total = 198 | % | Total = 76 | RR$_{crude}$ | 95% CI | p-value | | RR$_{adj}$ | 95% CI | p-value | |
| Antimalarials | 100.0 | 198 | 0.0 | 0 | – | – | – | | – | – | – | |
| Antibiotics | 23.7 | 47 | 15.8 | 12 | 0.7 | (0.4–1.1) | 0.2 | | 0.7 | (0.4–1.1) | 0.2 | |

Low-density (LD: sd-mRDT-/us-qPCR+) versus high-density *Pf* parasitemia (HD: sd-mRDT+/us-qPCR+).

HD, high-density *Pf* infection; LD, low-density *Pf* infection; RR, risk ratio; RR$_{adj}$; adjusted RR; RR$_{crude}$, crude (unadjusted) RR; sd-mRDT, standard malaria rapid diagnostic test; us-qPCR, ultrasensitive quantitative PCR.

[a]$p < 0.05$ and considered as statistically significant.

**Table 5. Comparative analysis of clinical presentation, lab results, and diagnoses at day zero.** LD versus ND (low-density versus undetectable)

| | ND (Undetectable) sd-mRDT-/us-qPCR- | | LD parasitemia sd-mRDT-/us-qPCR+ | | LD versus ND parasitemia | | | | | |
| --- | --- | --- | --- | --- | --- | --- | --- | --- | --- | --- |
| | Prevalence | | | | Crude bivariate analysis | | | Adjusted (season) | | |
| Signs and symptoms | % | Total = 2,527 | % | Total = 76 | RR$_{crude}$ | 95% CI | p-value | RR$_{adj}$ | 95% CI | p-value |
| Danger signs present | 1.3 | 34 | 1.3 | 1 | 1.0 | (0.1–6.1) | 1.0 | 1.4 | (0.2–8.1) | 0.8 |
| Respiratory distress | 7.1 | 180 | 1.3 | 1 | 0.2 | (0.0–1.2) | 0.08 | 0.4 | (0.1–2.6) | 0.3 |
| Cough | 59.1 | 1,493 | 39.5 | 30 | 0.5 | (0.3–0.7) | 0.001 [a] | 0.5 | (0.3–0.8) | 0.003 [a] |
| Pharyngitis | 1.4 | 36 | 1.3 | 1 | 0.9 | (0.1–5.8) | 0.9 | 1.1 | (0.1–6.6) | 0.9 |
| Abdominal pain | 4.1 | 103 | 6.6 | 5 | 1.6 | (0.7–3.8) | 0.3 | 1.5 | (0.6–3.6) | 0.4 |
| Loss of appetite | 2.3 | 58 | 3.9 | 3 | 1.7 | (0.5–5.0) | 0.4 | 1.9 | (0.6–5.4) | 0.3 |
| Vomit | 19.1 | 483 | 26.3 | 20 | 1.5 | (0.9–2.4) | 0.1 | 1.3 | (0.8–2.2) | 0.3 |
| Diarrhea | 17.3 | 438 | 14.5 | 11 | 0.8 | (0.4–1.5) | 0.5 | 0.8 | (0.4–1.4) | 0.4 |
| Fever only | 13.1 | 332 | 13.2 | 10 | 1.0 | (0.5–1.9) | 1.0 | 1.0 | (0.5–1.9) | 1.0 |
| FWS[b] | 29.8 | 753 | 43.4 | 33 | 1.8 | (1.1–2.7) | 0.01 [a] | 1.7 | (1.1–2.6) | 0.03 [a] |
| Lab results and measures | Mean | CI95% | Mean | CI95% | RR$_{crude}$ | 95% CI | p-value | RR$_{adj}$ | 95% CI | p-value |
| Temperature (°C) | 38.3 | (38.3–38.3) | 38.3 | (38.1–38.5) | 1.0 | (0.7–1.5) | 0.8 | 1.1 | (0.8–1.5) | 0.7 |
| Preconsult fever duration | 1.5 | (1.4–1.5) | 1.3 | (1.1–1.5) | 0.8 | (0.6–1.1) | 0.1 | 0.9 | (0.6–1.1) | 0.3 |
| Hemoglobin (g/dL) | 9.7 | (9.7–9.8) | 9.6 | (9.2–10.1) | 0.9 | (0.8–1.2) | 0.6 | 0.9 | (0.8–1.2) | 0.6 |
| | Median | IQR | Median | IQR | RR$_{crude}$ | 95% CI | p-value | RR$_{adj}$ | 95% CI | p-value |
| CRP (mg/L) | 0.0 | (0.0–10.0) | 0.0 | (0.0–10.0) | 1.0 | (1.0–1.0) | 0.7 | 1.0 | (1.0–1.0) | 0.9 |
| PCT (ug/L) | 0.2 | (0.1–0.6) | 0.3 | (0.1-0.4) | 0.9 | (0.8–1.1) | 0.5 | 0.9 | (0.8–1.1) | 0.4 |
| Diagnoses | % | Total = 2,527 | % | Total = 76 | RR$_{crude}$ | 95% CI | p-value | RR$_{adj}$ | 95% CI | p-value |
| Anemia (moderate-to-severe)[c] | 54.8 | 700/1,277 | 56.5 | 26/46 | 1.0 | (0.6–1.9) | 0.8 | 1.0 | (0.6–1.8) | 0.9 |
| Sickle cell disease (HbSS)[d] | 1.8 | 16/895 | 0.0 | 0/40 | 0.6 | (0.2–1.6) | 0.3 | 0.6 | (0.2–1.7) | 0.4 |
| Sickle cell trait (HbAS)[d] | 13.9 | 124/895 | 10.0 | 4/40 | | | | | | |
| Malnutrition | 7.2 | 181 | 9.2 | 7 | 1.3 | (0.6–2.7) | 0.5 | 1.4 | (0.6–2.9) | 0.4 |
| Severe malnutrition | 1.7 | 43 | 5.3 | 4 | 3.0 | (1.1–7.5) | 0.03 [a] | 3.0 | (1.1–7.5) | 0.03 [a] |
| Severe illness | 5.6 | 142 | 11.8 | 9 | 2.2 | (1.1–4.0) | 0.02 [a] | 2.6 | (1.3–4.8) | 0.01 [a] |
| URTI | 13.2 | 331/2,503 | 13.3 | 10 | 1.0 | (0.5–1.9) | 1.0 | 1.0 | (0.5–1.8) | 0.9 |
| Pneumonia | 26.8 | 678 | 13.2 | 10 | 0.4 | (0.2–0.8) | 0.01 [a] | 0.5 | (0.3–1.0) | 0.04 [a] |
| HIV | 1.4 | 33/2,311 | 1.5 | 1/68 | 1.0 | (0.1–6.4) | 1.0 | 1.6 | (0.2–9.3) | 0.6 |
| Suspected viral infection | 25.9 | 655 | 42.1 | 32 | 2.1 | (1.3–3.1) | 0.002 [a] | 1.9 | (1.2–2.9) | 0.01 [a] |
| Suspected bacterial infection | 4.0 | 102 | 1.3 | 1 | 0.3 | (0.1–2.2) | 0.3 | 0.3 | (0.1–2.3) | 0.3 |

[a] p < 0.05 and considered as statistically significant

[b] FWS is diagnosed using sd-mRDT, and it is thus not present in HD infection.

[c] Moderate-to-severe anemia: Hb < 9 g/dL.

[d] HbSS only investigated in samples with sufficient blood volume (i.e., 40/78 LD, 87/198 HD, and 895/2,527).

[e] CRP was measured using a categorical quantitation test. Four categories exist: <10 mg/L, 10–40 mg/L, 40–80 mg/L, and >80 mg/L. The lower limit of this range is represented in the table. See S2 Table for the definition of severe illness and severe malnutrition.

CRP, C-reactive protein; FWS, fever without source; HbAS heterozygote, sickle cell trait; HbSS homozygote, sickle cell disease; HD, low-density *Pf* infection; HIV, human immunodeficiency virus; LD, low-density *Pf* infection; PCT, procalcitonin; RR, risk ratio; RR$_{adj}$, adjusted RR; RR$_{crude}$, crude (unadjusted) RR; Sd-mRDT, standard malaria rapid diagnostic test; URTI, upper respiratory tract infection; us-qPCR, ultrasensitive quantitative PCR.

LD infections were 3-fold more likely to occur in severely malnourished children when compared with ND (95% CI 1.1–7.5, *p* = 0.03, Table 5), a difference not found between HD and LD groups (RR = 1.6, 95% CI 0.5–2.9, *p* = 0.3, Table 6). This explains the higher prevalence of "severe illnesses" in LD compared with ND (RR = 2.2, 95% CI 1.1–4.0, *p* = 0.02), an observation which became statistically insignificant when controlling for severe malnutrition

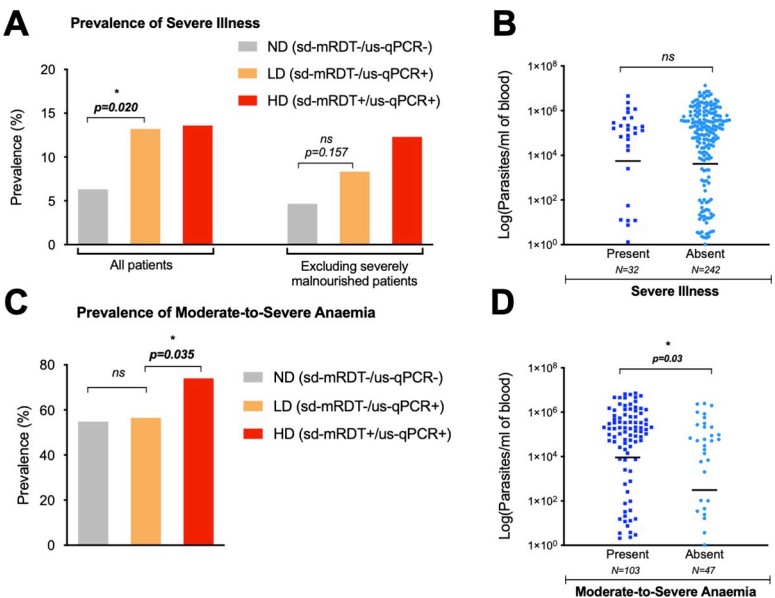

**Fig 3. Distribution of severe illness and anemia between patients with LD, HD, or ND parasitemia. (A)** Prevalence of severe illness amongst all patients (left) versus a subgroup excluding severely malnourished patients (right). **(B)** Log parasite densities in *Pf*-positive cases (qPCR-detectable: LD and HD), in the presence or absence of severe illness. **(C)** Prevalence of patients with moderate-to-severe anemia. **(D)** Log parasite densities in *Pf*-positive cases (qPCR-detectable: LD and HD), in the presence or absence of moderate-to-severe anemia. Severe malnutrition is defined as weight-for-age $Z$-score $< -3$ and/or MUAC $<5$–$11$ cm. Moderate-to-severe anemia is defined as memoglobin $<9$ g/dl. *$p < 0.05$. HD, high-density *Pf* infection, LD, low-density *Pf* infection; RDT, rapid diagnostic test; MUAC, mid-upper arm circumference; ND, no detectable *Pf* parasitemia; ns, nonsignificant; sd-mRDT, standard malaria RDT; us-qPCR, ultrasensitive quantitative polymerase chain reaction.

(RR = 1.8, 95% CI 0.8–4.0, $p = 0.2$) (**Fig 3A, Table 5**). Indeed, among ultrasensitive qPCR+ patients, no trend between parasite density and illness severity was observed, and mean parasitemia was not different between those with severe and nonsevere illnesses (**Fig 3B**).

**LD versus HD.** Only markers of anemia and inflammation were significantly elevated in HD infections, **Table 6**). For instance, HD infections had significantly higher axillary temperatures compared with LD, (39.0˚C versus 38.3˚C, RR = 2.5, 95% CI 2.0–3.3, $p < 0.001$), whereas no differences were observed between LD and ND groups (**Table 5**). Additionally, moderate-to-severe anemia ($<9$ g/dl) was 3-fold more frequent in the HD group (RR = 3.3, 95% CI 1.4–10.0, $p < 0.001$) (**Fig 3C**) and corresponded to a mean difference of 0.7 g/dl hemoglobin (LD: 9.6 versus HD: 8.9 g/dl, 95% CI 0.2–1.4) (**Table 6**). This trend was also visible at the parasitemia level, in which children with moderate-to-severe anemia ($<9$ g/dl) had significantly higher parasite loads ($5.1 \times 10^5$ more parasites, 95% CI $5.4 \times 10^4$ to $9.6 \times 10^5$, $p = 0.03$) (**Fig 3D**). No differences were seen in sickle trait carriage or disease across any group (**Table 6**). Amongst inflammatory markers, children with HD infection had 17 ug/L higher mean procalcitonin (medians 0.3 ug/L versus 5.5 ug/L, RR = 1.4, 95% CI 1.0–2.0, $p = 0.04$) (**Fig 4A**) and 13.2 mg/L higher mean CRP (medians of a range of 0–10 mg/L versus 10–40 mg/L, $p = 0.02$) (**Fig 4B**).

## Clinical outcome of children with *Pf* parasitemia

**Clinical outcomes (LD versus ND).** Despite the fact that children with LD were not treated with antimalarials, neither the proportion of clinical failures (RR = 0.7, 95% CI 0.2–2.6, $p = 0.5$) (**Fig 5A**) nor the proportion of secondary hospitalizations (RR = 0.9, 95% CI 0.2–3.5, $p = 0.9$) (**Fig 5B**) were significantly different between LD and ND groups (**Table 7**). The

**Table 6. Comparative analysis of clinical presentation, lab results and diagnoses at day zero.** LD versus HD (low versus high-density parasitemia).

| | HD parasitemia sd-mRDT+/us-qPCR+ | | LD parasitemia sd-mRDT-/us-qPCR+ | | LD versus HD parasitemia | | | | | |
|---|---|---|---|---|---|---|---|---|---|---|
| | PREVALENCE | | | | CRUDE BIVARIATE ANALYSIS | | | ADJUSTED (Age) | | |
| Signs and Symptoms | % | Total = 198 | % | Total = 76 | $RR_{crude}$ | 95% CI | p-value | $RR_{adj}$ | 95% CI | p-value |
| Danger signs present | 2.0 | 4 | 1.3 | 1 | 0.7 | (0.1–2.4) | 0.7 | 0.4 | (0.1–2.1) | 0.4 |
| Respiratory distress | 0.5 | 1 | 1.3 | 1 | 1.8 | (0.2–3.4) | 0.5 | 1.3 | (0.1–3.2) | 0.8 |
| Cough | 44.4 | 88 | 39.5 | 30 | 0.9 | (0.6–1.3) | 0.5 | 0.7 | (0.5–1.1) | 0.2 |
| Pharyngitis | 1.0 | 2 | 1.3 | 1 | 1.2 | (0.2–3.0) | 0.8 | 1.0 | (0.1–2.9) | 1.0 |
| Abdominal pain | 12.6 | 25 | 6.6 | 5 | 0.6 | (0.2–1.2) | 0.2 | 1.1 | (0.5–2.0) | 0.9 |
| Loss of appetite | 2.5 | 5 | 3.9 | 3 | 1.6 | (0.4–2.6) | 0.5 | 1.3 | (0.4–2.7) | 0.6 |
| Vomit | 24.2 | 48 | 26.3 | 20 | 1.1 | (0.7–1.6) | 0.7 | 1.0 | (0.6–1.5) | 0.9 |
| Diarrhea | 8.1 | 16 | 14.5 | 11 | 1.5 | (0.9–2.3) | 0.12 | 1.3 | (0.7–2.1) | 0.4 |
| Fever only | 23.2 | 46 | 13.2 | 10 | 0.6 | (0.3–1.0) | 0.07 ~ | 0.6 | (0.3–1.1) | 0.09 ~ |
| FWS[b] | – | – | 43.4 | 33 | – | – | – | – | – | – |
| Lab results and measures | Mean | 95% CI | Mean | 95% CI | $RR_{crude}$ | 95% CI | p-value | $RR_{adj}$ | 95% CI | p-value |
| Temperature (°C) | 39.0 | (38.8–39.1) | 38.3 | (38.1–38.5) | 0.4 | (0.3–0.5) | <0.001 a | 0.4 | (0.3–0.6) | <0.001 a |
| Preconsult fever duration | 1.7 | (1.5–1.8) | 1.3 | (1.1–1.5) | 0.7 | (0.6–1.0) | 0.02 a | 0.7 | (0.6–1.0) | 0.06 ~ |
| Hemoglobin (g/dL) | 8.9 | (8.5–9.2) | 9.6 | (9.2–10.1) | 1.3 | (1.0–1.6) | 0.02 a | 1.6 | (1.2–2.1) | 0.00 a |
| | Median | IQR | Median | IQR | $RR_{crude}$ | 95% CI | p-value | $RR_{adj}$ | 95% CI | p-value |
| CRP (mg/L)[e] | 10.0 | (10.0–40.0) | 0.0 | (0.0–10.0) | 1.0 | (0.9–1.0) | 0.02 a | 1.0 | (0.9–1.0) | 0.05 a |
| PCT (ug/L) | 5.5 | (0.6–21.1) | 0.3 | (0.1-0.4) | 0.7 | (0.5–1.0) | 0.04 a | 0.7 | (0.5–1.0) | 0.07 ~ |
| Diagnoses | % | Total = 198 | % | Total = 76 | $RR_{crude}$ | 95% CI | p-value | $RR_{adj}$ | 95% CI | p-value |
| Anemia (moderate-to-severe)[c] | 74.0 | 77/104 | 56.5 | 26/46 | 0.6 | (0.3–1.0) | 0.04 a | 0.3 | (0.1–0.7) | 0.001 a |
| Sickle cell disease (HbSS)[d] | 14.9 | 13/87 | 10.0 | 4/40 | 0.7 | (0.3–1.5) | 0.5 | 0.7 | (0.3–1.6) | 0.5 |
| Sickle cell trait (HbAS)[d] | 0.0 | 0/87 | 0.0 | 0/40 | | | | | | |
| Malnutrition | 5.6 | 11 | 9.2 | 7 | 1.4 | (0.7–2.3) | 0.3 | 1.2 | (0.6–2.1) | 0.6 |
| Severe malnutrition | 1.5 | 3 | 5.3 | 4 | 2.1 | (0.8–3.1) | 0.1 | 1.6 | (0.5–2.9) | 0.3 |
| Severe illness | 11.6 | 23 | 11.8 | 9 | 1.0 | (0.5–1.7) | 1.0 | 1.1 | (0.6–1.8) | 0.8 |
| URTI | 5.6 | 11 | 13.3 | 10/75 | 1.8 | (1.0–2.7) | 0.04 a | 1.5 | (0.8–2.4) | 0.2 |
| Pneumonia | 23.7 | 47 | 13.2 | 10 | 0.6 | (0.3–1.0) | 0.06 ~ | 0.6 | (0.3–1.1) | 0.08 ~ |
| HIV | 0.5 | 1/185 | 1.5 | 1/68 | 1.9 | (0.2–3.5) | 0.5 | 1.7 | (0.2–3.5) | 0.6 |
| Suspected viral infection | 0.5 | 1 | 42.1 | 32 | 5.3 | (4.4–5.5) | <0.001 a | 5.3 | (4.4–5.5) | <0.001 a |
| Suspected bacterial infection | 0.5 | 1 | 1.3 | 1 | 1.8 | (0.2–3.4) | 0.5 | 2.5 | (0.3–3.6) | 0.3 |

[a]$p < 0.05$ and considered as statistically significant

[b]FWS is diagnosed using sd-mRDT, and it is thus not present in HD infection.

[c]Moderate-to-severe anemia: Hb < 9 g/dL.

[d]HbSS only investigated in samples with sufficient blood volume (i.e., 40/78 LD, 87/198 HD, and 895/2,527).

[e]CRP was measured using a categorical quantitation test. Four categories exist: <10 mg/L, 10–40 mg/L, 40–80 mg/L, and >80 mg/L. The lower limit of this range is represented in the table. See S2 Table for the definition of severe illness and severe malnutrition.

CRP, C-reactive protein; FWS, fever without source; HbAS heterozygote, sickle cell trait; HbSS homozygote, sickle cell disease; HD, low-density *Pf* infection; HIV, human immunodeficiency virus; LD, low-density *Pf* infection; PCT, procalcitonin; RR, risk ratio; $RR_{adj}$, adjusted RR; $RR_{crude}$, crude (unadjusted) RR; Sd-mRDT, standard malaria rapid diagnostic test; URTI, upper respiratory tract infection; us-qPCR, ultrasensitive quantitative PCR.

frequency of deaths was zero in LD children and or near zero (0.3%, *n* = 8/2,527) in ND children (Table 7). No significant differences were observed in secondary outcomes, such as the number of days until fever clearance (postconsultation) and the duration of secondary hospital admissions. Finally, none of the 76 children with LD infections tested positive for HD *Pf* parasitemia in the first 7-day follow-up period despite not receiving antimalarials.

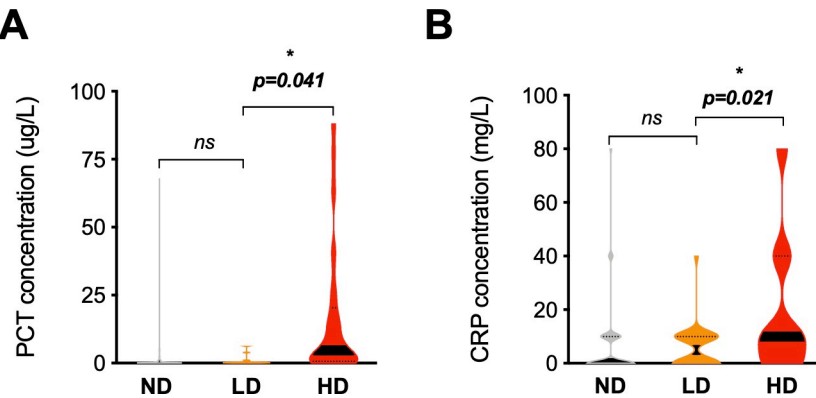

**Fig 4. Distribution of inflammatory markers between patients with LD, HD, or ND parasitemia.** Violin plots showing the distribution in the concentrations of (**A**) PCT and (**B**) CRP found among ND, LD, and HD groups. Thick black bands: median. Dotted lines: quartiles. CRP was measured using a categorical quantitation test. Four categories exist: <10 mg/L, 10–40 mg/L, 40–80 mg/L, and >80 mg/L. The lower limit of this range is represented in the plot. *$p < 0.05$. CRP, C-reactive protein; HD, high-density *Pf* infection; LD, low-density *Pf* infection; ND, undetectable *Pf* parasitemia; ns, nonsignificant; PCT, procalcitonin; Sd-mRDT, standard malaria rapid diagnostic test; us-qPCR, ultrasensitive quantitative PCR.

**Clinical outcomes (LD versus HD).** In contrast, children with HD infections were 3.3-fold more likely to be admitted to hospital during the 28 days of follow-up (95% CI 0.1–1.0, $p = 0.06$) when compared to LD (**Fig 5B**). No differences were observed in the duration of secondary hospital admissions nor in the time to fever clearance (**Table 8**).

## Discussion

This study explores the clinical consequences of using highly sensitive diagnostic tools (ultrasensitive qPCR and ultrasensitive mRDT) for the detection of LD *Pf* parasitemia amongst febrile pediatric outpatients in a moderate endemicity setting. In this cohort of 2,801 Tanzanian children, the prevalence of *Pf* parasitemia was 9.8% by ultrasensitive qPCR. A quarter of these infections were not detected by both standard mRDTs and ultrasensitive mRDTs. The performance of the ultrasensitive-mRDT was previously tested on this cohort, in which it was

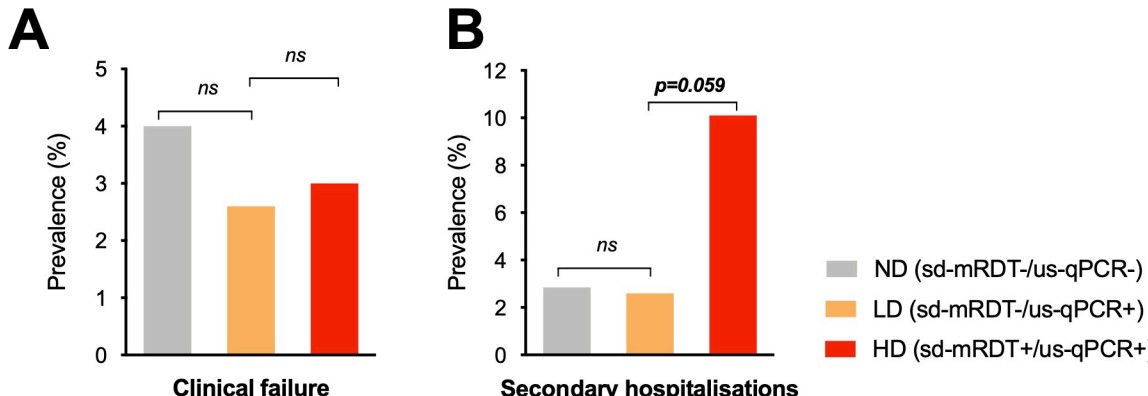

**Fig 5. Distribution of clinical outcomes between patients with LD, HD, or ND parasitemia.** (**A**) Prevalence of clinical failure (patients not cured) by day 7 postconsultation, (**B**) Prevalence of patients requiring secondary hospital admission. See **S2 Table** for the definition of severe illness. *$p < 0.05$. HD, high-density *Pf* infection; LD, low-density *Pf* infection; ND, undetectable *Pf* parasitemia; ns, nonsignificant; PCT, procalcitonin; sd-mRDT, standard malaria rapid diagnostic test; us-qPCR, ultrasensitive quantitative PCR.

**Table 7.  Comparative analysis of health outcomes.**

| | ND sd-mRDT-/us-qPCR- | | LD parasitemia sd-mRDT-/us-qPCR+ | | LD versus ND parasitemia | | | | | |
| --- | --- | --- | --- | --- | --- | --- | --- | --- | --- | --- |
| | Prevalence | | | | Crude bivariate analysis | | | Adjusted (season) | | |
| Outcomes | Mean | 95% CI | Mean | 95% CI | RR$_{crude}$ | 95% CI | p-value | RRadj | 95% CI | p-value |
| Days to fever clearance | 3.5 | (3.4–3.5) | 3.2 | (2.8–3.5) | 0.9 | (0.8 to −1.0) | 0.2 | 0.9 | (0.8–1.1) | 0.4 |
| Admission duration (days) | 5.1 | (4.1–6.0) | 3.0 | (−22.4 to 28.4) | 0.8 | (0.4–1.6) | 0.5 | 0.7 | (0.3–1.5) | 0.4 |
| | % | Total = 2,527 | % | Total = 76 | | 95% CI | p-value | RRadj | CI95% | p-value |
| Clinical failure | 4.0 | 101 | 2.6 | 2 | 0.7 | (0.2–2.6) | 0.6 | 0.9 | (0.2–3.5) | 0.9 |
| Secondary hospitalization | 2.8 | 72 | 2.6 | 2 | 0.9 | (0.2–3.5) | 0.9 | 1.0 | (0.2–4.1) | 0.9 |
| Death | 0.3 | 8 | 0.0 | 0 | – | – | – | – | – | – |

LD versus ND (low-density versus undetectable *Pf* parasitemia).

See S2 Table for the definition of clinical failure.

HD, high-density *Pf* infection; LD, low-density P*f* infection; RR, risk ratio; RR$_{adj}$, adjusted RR. sd-mRDT, standard malaria RDT; us-qPCR, quantitative PCR.

also described as having near identical sensitivity and specificity to standard mRDTs [14]. The children with LD infections did not have a significantly different clinical outcome compared with *Pf*-negative children, despite not having received antimalarial treatment.

It was previously thought that LD infections and asymptomatic carriage mostly occurred in high-endemicity areas because of a higher immune tolerance and control. However, it is becoming clear that these infections are even more common in regions with low malaria endemicity after a recent period of high transmission [17]. This trend raises the question of whether LD infections have evolved as a survival strategy to maintain parasite transmission [18]. Indeed, these persistent, slowly oscillating low parasite densities may persist in untreated individuals for several months to years [19]. Interestingly, in our study, LD infection prevalence fell to zero during the non-malaria season despite continued testing, perhaps an indication that LD infection was not functioning as an off-season reservoir. Larger epidemiological surveys are required before such conclusions can be made. In this context, LD carriage may rather represent passing self-resolving *Pf* infections, which are more or less rapidly cleared by children according to their level of immunity (children with severe malnutrition had indeed a much higher rate of LD infections).

**Table 8.  Comparative analysis of health outcomes.**

| | HD parasitemia sd-mRDT+/us-qPCR+ | | LD parasitemia sd-mRDT-/us-qPCR+ | | LD versus HD parasitemia | | | | | |
| --- | --- | --- | --- | --- | --- | --- | --- | --- | --- | --- |
| | Prevalence | | | | Crude bivariate analysis | | | Adjusted (age) | | |
| Outcomes | Mean | 95% CI | Mean | 95% CI | RR$_{Crude}$ | 95% CI | p-value | RR$_{adj}$ | 95% CI | p-value |
| Days to fever clearance | 3.5 | (3.2–3.7) | 3.2 | (2.8–3.5) | 0.9 | (0.7–1.0) | 0.1 | 0.9 | (0.7–1.0) | 0.1 |
| Admission duration (days) | 3.6 | (3.0–4.2) | 3.0 | (−22.4 to 28.4) | 0.7 | (0.2–2.3) | 0.5 | 0.5 | (0.1–2.1) | 0.4 |
| | % | Total = 198 | % | Total = 76 | | 95% CI | p-value | RR$_{adj}$ | 95% CI | p-value |
| Clinical failure | 3.0 | 6 | 2.6 | 2 | 0.9 | (0.2–4.4) | 0.9 | 0.9 | (0.2–4.9) | 0.9 |
| Secondary hospitalization | 10.1 | 20 | 2.6 | 2 | 0.3 | (0.1–1.0) | 0.1 ~ | 0.3 | (0.1–1.0) | 0.06 ~ |
| Death | 0.0 | 0 | 0.0 | 0 | – | – | – | – | – | – |

LD versus HD (low-density versus high-density *Pf* parasitemia).

See S2 Table for the definition of clinical failure.

HD, high-density *Pf* infection; LD, low-density P*f* infection; RR, risk ratio; RR$_{adj}$, adjusted RR. sd-mRDT, standard malaria RDT; us-qPCR, quantitative PCR.

The mechanism behind the long-term asymptomatic maintenance of LD infections is thought to rely on immunological tolerance, in which the presence of neutralizing antibodies is postulated to keep parasitemia at low levels [18]. This could also explain the age trend observed in our study, in which parasitemia followed a directly proportional association with age, and the peak probability of LD infections occurred in infants under 9 months old who are under the protection of maternal passive immunity. In malaria-endemic countries, this age group has a significantly lower risk of developing severe malaria [20, 21]. This natural resistance in infants is likely to rather rely on passive immunity maternal antibodies, which impair the cytoadherence of parasitized red blood cells [22]. Further study into the immunological mechanism of LD carriage is needed before conclusions can be drawn on the origin of infant resistance to severe malaria.

As expected, the presence and severity of anemia had a directly proportional relationship with parasitemia amongst ultrasensitive qPCR+ individuals. However, although moderate-to-severe anemia (<9 g/dl) was significantly more frequent in HD compared with either LD or ND, no differences in hemoglobin levels were found between LD and ND groups. Sickle cell anemia is known to be protective of HD *Pf* parasitemia, and previous studies have revealed that these patients are also predisposed to LD carriage [23]. In this study, however, no differences in sickle cell disease or trait were found among any groups, albeit that the analyses were limited by the low prevalence of the HbSS trait. Thus, no associations between anemia and LD carriage were found within the follow-up timeframe of this cohort.

As exposure to *Pf* antigens is essential to transform passive maternal immunity into an active memory response, an argument can be made that asymptomatic LD carriage in children may be beneficial in the long term [24, 25]. Indeed, it has been previously shown that frequent superinfections (i.e., a high number of concurrent clones) can be protective against clinical malaria [26]. A counterargument is that the long-term stress of the infection may be associated with increased chronic morbidity and all-cause mortality [27]. However, these events are not measurable at the outpatient level or in the timeframe of acute febrile disease. Following this, our study found no statistically significant clinical impact of untreated LD parasitemia when compared with febrile controls without detectable *Pf* parasitemia (ND). In contrast, untreated patients with LD infections fared significantly better than those with HD infections (who were treated with antimalarials). For instance, despite treatment, HD infections had a significantly increased risk of developing severe outcomes, necessitating secondary hospitalization over the 28-day follow-up period. Critically, no differences were seen in the number of clinical failures at day 7 when the LD group was compared with either HD or ND, despite the fact that LD patients did not receive antimalarial therapy. This indicates that these LD infections represented either self-resolving fevers or an incidental finding in children with an alternative (most often viral) infection. However, as the pyrogenic threshold of parasitemia has been previously described to fall well within standard mRDT limits of detection, it is unlikely that LD infection was the primary cause of the febrile episode and the possibility of synergistic pyrogenic interactions with coinfecting viruses could also be possible [28]. Indeed, children with LD infections were significantly more likely to have suspected viral infections compared with either HD or ND groups according to their clinical presentation (undifferentiated fever), low level CRP and PCT (which are commonly associated with bacterial infections when high) and good outcome without antibiotic treatment. The bacteremia and sepsis in this study almost exclusively occurred in *Pf*-negative children (93%; *n* = 14/15).

Another question raised by these findings is whether LD infections are actually just clinically precocious HD infections. However, none of the children with LD infections had persistent fever that later tested positive for HD *Pf* parasitemia during the first 7 days of follow-up, despite not receiving antimalarials. Importantly, the data of this study were collected using

electronic clinical decision support algorithms, which guided clinicians to appropriate management strategies and diagnoses. This situation differs significantly from routine care, in which clinicians are much more likely to diagnose and treat malaria regardless of test results. In this situation, the use of overly sensitive malaria diagnostic tests would likely result in even more overprescriptions and, possibly, missed alternate diagnoses. However, the clinical impact of these potentially missed diagnoses would be limited as the vast majority of these LD children were suffering from presumed viral infections, which resolved by themselves without antibiotic treatment.

## Limitations and further research

As this cohort only included children between 2 and 59 months of age, it is not generalizable to adults or neonates who have different immunological characteristics. Further, as all the patients were febrile, the association to fever itself cannot be drawn. More research is required in more geographically and clinically diverse environments (including community-based studies and healthy controls) that will provide the statistical capacity to assess the aparasitemic cut-offs for *Pf* parasite-attributable morbidity. Longer-term follow-up models will also help investigate the potential importance of parasite carriage and recurrent infections. Finally, this study does not investigate causality of LD parasitemia, and thus cannot answer what immunologic, environmental, or genetic factors (of the host, parasite, or vector) may cause a person to resist infections progressing to higher parasitemia. The multifactorial possibilities merit multiple in depth studies. Of particular interest may be an investigation into the effect of coinfections, which may have an immunomodulatory influence on the control *Pf* coinfections, such as has been suggested for the gut microbiome [29]. For example, there is specific evidence that viral infections such as influenza and measles may reduce parasite density in febrile children, and this poses an potentially important research question during the outbreak caused by severe acute respiratory syndrome coronavirus 2 (SARS-CoV-2) [30].

## Impact and recommendations

Currently, WHO uses 2 case definitions for malaria depending on the level of transmission: In areas with moderate-to-high transmission where "malaria-control" is the aim, cases are defined as any symptomatic person with (any) positive diagnostic test. However, no symptoms are necessary to define malaria in "malaria elimination" settings [31]. In a clinical context, not having a parasitemic cut-off for the definition of malaria may cause significant confusion. As clinically inconsequential carriage may occur alongside other serious diseases, the use of ultra-sensitive diagnostics to detect LD *Pf* parasitemia may create red herring signals that mislead the clinician to overlook the true underlying diagnosis if not well trained. Thus, even though our study did not show a risk of missing a serious infection, diagnostic sensitivity of parasite detection must still be adapted to clinical relevance (i.e., there is a need to define parasitemic thresholds to guide clinicians on whether the detected parasites are likely to be symptomatic and/or a risk for community transmission). More importantly, training and guidance to clinicians should be provided to help manage febrile patients in an integrated way, through the provision of (electronic) evidence-based algorithms and guidelines [32]. Indeed, in this study, clinicians strictly followed electronic algorithms, which are not yet widely available elsewhere and may have helped avoid missing serious infections.

Thus, it is important to appreciate that the need for administering antimalarial therapy is proportional to parasite density. LD *Pf* parasitemia infections must be targeted to achieve elimination [33], but our study shows that there is no short-term consequence to withholding antimalarials from acutely febrile individuals with LD infection in the context and timeframe of an

outpatient setting. These results bolster several robust studies, which conclude that it is safe to use only standard mRDTs in malarial diagnosis as is recommended in the current WHO guidelines [10–12, 34].

## Conclusion

In this study in a moderate malaria endemicity setting, LD *Pf* parasitemia infections undetected by mRDT did not present acute health concerns in children in an outpatient setting. These LD infections may thus either represent benign self-resolving fevers or an incidental finding in children with other infections, likely of viral origin and thus requiring no antibiotic treatment. Our findings suggest that implementing ultrasensitive malaria diagnostics to detect LD infections is neither useful nor deleterious (except for wasting resources) in this context.

## Supporting information

**S1 STROBE checklist. STROBE, Strengthening the Reporting of Observational Studies in Epidemiology.**
(DOCX)

**S1 Table. Results of standard mRDT and ultrasensitive mRDT stratified by ultrasensitive qPCR.** HD, high-density *Pf* infection; LD, low-density *Pf* infection; ND, no detectable *Pf* parasitemia; sd-mRDT, standard malaria RDT; us-mRDT, ultrasensitive-mRDT (Alere); us-qPCR, ultrasensitive quantitative PCR.
(DOCX)

**S2 Table. Definition of clinical failure (primary outcome measure).** Adapted from Keitel et al. [13].
(DOCX)

## Acknowledgments

We thank the patients and caretakers involved in this study for their contribution to work toward improving the diagnosis of fever in children.

## Author Contributions

**Conceptualization:** Mary-Anne Hartley, Blaise Genton, Valérie D'Acremont.

**Data curation:** Mary-Anne Hartley, Natalie Hofmann, Kristina Keitel, Frank Kagoro, Clara Antunes Moniz, Tarsis Mlaganile, Josephine Samaka, John Masimba, Zamzam Said, Hosiana Temba, Iveth Gonzalez.

**Formal analysis:** Mary-Anne Hartley, Natalie Hofmann.

**Funding acquisition:** Ingrid Felger, Valérie D'Acremont.

**Investigation:** Mary-Anne Hartley, Natalie Hofmann, Frank Kagoro, Clara Antunes Moniz, Tarsis Mlaganile, Josephine Samaka, John Masimba, Zamzam Said, Hosiana Temba, Iveth Gonzalez.

**Methodology:** Mary-Anne Hartley.

**Project administration:** Mary-Anne Hartley, Blaise Genton, Valérie D'Acremont.

**Supervision:** Kristina Keitel, Ingrid Felger, Blaise Genton, Valérie D'Acremont.

**Validation:** Mary-Anne Hartley, Natalie Hofmann, Kristina Keitel, Iveth Gonzalez, Ingrid Felger.

**Visualization:** Mary-Anne Hartley.

**Writing – original draft:** Mary-Anne Hartley.

**Writing – review & editing:** Mary-Anne Hartley, Natalie Hofmann, Kristina Keitel, Frank Kagoro, Ingrid Felger, Blaise Genton, Valérie D'Acremont.

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
