## [Editor Report · Decision Letter 0]

17 Feb 2020

Dear Dr Hartley, 

Thank you for submitting your manuscript entitled "Clinical relevance of low-density Plasmodium falciparum parasitaemia in untreated febrile children: a retrospective cohort analysis" for consideration by PLOS Medicine.

Your manuscript has now been evaluated by the PLOS Medicine editorial staff [as well as by an academic editor with relevant expertise] and I am writing to let you know that we would like to send your submission out for external peer review.

Kind regards,

Clare Stone, PhD,

PLOS Medicine

---

## [Decision Letter · Decision Letter 1]

26 May 2020

Dear Dr. Hartley,

Thank you very much for submitting your manuscript "Clinical relevance of low-density Plasmodium falciparum parasitaemia in untreated febrile children: a retrospective cohort analysis." (PMEDICINE-D-20-00426R1) for consideration at PLOS Medicine. 

[LINK]

In light of these reviews, I am afraid that we will not be able to accept the manuscript for publication in the journal in its current form, but we would like to consider a revised version that addresses the reviewers' and editors' comments. Obviously we cannot make any decision about publication until we have seen the revised manuscript and your response, and we plan to seek re-review by one or more of the reviewers. 

We expect to receive your revised manuscript by Jun 16 2020 11:59PM. Please email us (plosmedicine@plos.org) if you have any questions or concerns.

We look forward to receiving your revised manuscript. 

Sincerely,

Emma Veitch, PhD

PLOS Medicine

On behalf of Clare Stone, PhD, Acting Chief Editor,

PLOS Medicine

plosmedicine.org

*Currently the title has been phrased per the usual PLOS Medicine style, which is good (study question followed by the type of study design after a colon) but one reviewer questions whether it was really a retrospective analysis - the authors could consider this point and rephrase as needed.

*In the abstract conclusions, it would be good to take on board the reviewers' points (particularly reviewer 1) that the main finding of the analysis (failure to see a difference in clinical outcomes between LD and ND children) may have been presented too strongly, ie as a negative effect rather than failure to see an effect. This distinction could be made more clearly.

*In the last sentence of the Abstract Methods and Findings section, please describe briefly any key limitation(s) of the study's methodology.

*At this stage, we ask that you include a short, non-technical Author Summary of your research to make findings accessible to a wide audience that includes both scientists and non-scientists. The Author Summary should immediately follow the Abstract in your revised manuscript. This text is subject to editorial change and should be distinct from the scientific abstract. Please see our author guidelines for more information: https://journals.plos.org/plosmedicine/s/revising-your-manuscript#loc-author-summary

*In Materials and Methods, Design/Patient population and context section, this describes the work as a retrospective analysis of a prospective cohort study. However later on it's presented as an analysis based on patients enrolled in a randomized trial. I think this distinction (whether it was a cohort or a RCT) is likely to be confusing to readers, it may be clearer to say that this is an analysis using data from a RCT but combining data from the trial arms and therefore treating it like a cohort? Assume that this is indeed what was done? 

*We'd suggest also clarifying if the analysis presented here was prospectively planned out in a protocol or analysis plan? Please state this (either way) early in the Methods section.

*As noted by one reviewer, there are a few slightly ungrammatical phrases in the paper, I've suggested a few examples below. It may be worth cross-reading the paper before resubmitting to try to tidy some of these up. 

*Abstract - "The App collected at the same time systematic clinical data (symptoms, signs and laboratory tests on day 0 and at follow-up visits" - perhaps "The app SIMULTANEOUSLY collected clinical data ..."

*Abstract - "are used to propose (and record) a diagnosis and management plan" - in the paper "propose" is used in a number of places where in English, "suggest" or "put forward" might read more naturally. 

*Abstract - "Concerning malaria, only sd-mRDT+ cases received antimalarials" - "Only sd-mRDT+ cases received antimalarials.."

Comments from the reviewers:

Reviewer #1: See attachment

Michael Dewey

Reviewer #2: This is a well performed and well analysed study and a well written article. It adresses a clinically quite important subject/research question, namely the potential benefit vs even negative impact of using the recently developed ultrasensitive mRDTs vs the standard mRDTs - the article provides a rather credible, clear and important answer to this primary research question

The article also discusses a few general issues related to parasite densities of Pf infections such as maternal antibodies, sickle cell trait and concomitant infections - which I find adds to my interest in the article

Minor comments

a) I believe in Title of Table 1 = "sub-cohort" should be ""sub-cohorts"

b) maybe add one ref in rel to the text on withholding malaria treatment to mRDT neg febrile children in malaria endemic countries = Msellem et al PLoS Med 6:e 1000070 , 2009

c)definitely consider adding one ref on the effect of viral infections reducing malaria parasite densities in febrile Children = Rooth and Björkman Am J Trop Med 47: 675 , 1992 - this supports the findings in the present article

Reviewer #3: This manuscript indicates that failure to treat febrile P. falciparum infection with antimalarial drugs in Tanzanian children under 5 years with infections detectable by ultrasensitive PCR but not by standard RDT has no negative health consequences. The authors conclude that ultrasensitive diagnostics for Pf infection offer no clinical benefit in an urban primary care setting where malaria endemicity is low. (Dar es Salaam in this case.) 

Major comments

1) Strengths of the manuscript include: a) utilization of an electronic algorithm for clinical decision making that includes syndromic features and point of care diagnostics that are affordable in resource constrained settings, i.e., Keitel et al PLoS Med 2017; b) Comparison of Pf infections detected by ultrasensitive PCR to those detected by ultrasensitive RDT; c) The 28 day follow up period lends credibility to the claim that lack of administration of antimalarials in children who are positive for Pf by ultrasensitive PCR but negative by standard RDT (for PfHRP2) is appropriate; d) The study is clearly ethical as studies involving diagnosis by ultrasensitive PCR were done retrospectively.

2) Terminology in the manuscript with respect to "high density", "low density" parasiteimia, etc. is confusing and potentially misleading to some readers. This is particularly true in settings in sub-Saharan Africa where the Pf biomass is readily detectable by microscopic inspection of blood smears and RDTs are not routinely performed in order to reduce cost.

3) Severe malnutrition is a major confounder in the analysis. More detail on the criteria for the diagnosis of "severe malnutrition" in the ePOCT algorithm is needed.

4) Statements with respect to "severe illness" and anaemia as "moderate to severe" (as in Table 4A and 4B) are confusing and differ from those in the PLoS Med 2017 paper. Hb concentration values in the manuscript are given as mmol/L whereas values in the 2017 paper are presented as g/L. Secondly, the PLoS Med paper (Fig. 2) defines severe anemia as Hb <60 g/L (Hb 6 g/dL or 3.72 mmol/L). The high density parasitemia patients in this manuscript had a median value of 8.9 mmol/L (14.3 g/dL) and referred to as "severe anaemia". Please clarify what is meant by severe anaemia and what cut off values are used in this urban population. Attention to this detail is particularly important to the Discussion (e.g., lines 381-389).

Minor comments

1) Line 11. Capital letters are omitted from the institutional unit. 

2) Acronyms are over used. RDT and PCR are generally understood terms but readability would be greatly improved by avoiding terms such LD, ND, us-qPCR+ vs. qPCR- etc.

3) Table 1 is unnecessary and could be omitted. 

Reviewer #4: 

The paper is written in a rather unconventional way. Nothing wrong with innovation but I have to admit I found it hard to find my way around this paper. More confusion comes from near complete similarity of standard RDT+ and us-RDT+ study participants. Since the difference in sensitivity between the two RDTs is minimal, despite the name "ultrasensitive" there is very little gained from testing with us-RDTs. The us-RDTs missed the majority of the low density infections detected by PCR. Hence the inclusion of a patient group "us-RDT+" adds no information but adds considerable confusion. Would the authors consider omitting the data on us-RDTs since it contributes nothing meaningful to description of clinical sequelae of low density infections? I understand the desire to publish the data but the inclusion of the usRDT data do not make the paper stronger. 

Abstract

Provide total numbers included in each of the two groups: sd-mRDT-/us-qPCR+, sd-mRDT-/us-qPCR-, sd-mRDT+/us-qPCR+.

"Pf positivity rate was 7.1% and 9.8% by sd-mRDT and us-qPCR respectively." What % of usRDTs were positive?

Intro

"A new easy-to-use and affordable ultra-sensitive mRDT (us-mRDT) was developed (Alere™, Abbott Diagnostics) able to detect the Pf Histidine-Rich Protein-2 (HRP-2) with ten-fold greater sensitivity to that of sd-mRDTs." I do not understand why the authors use reference #8 if their own group has made much more realistic estimates of the added sensitivity of us-RDT? "us-RDT identified few additional P. falciparum-positive patients as compared to co-RDT (276 vs 265 parasite-positive patients detected), with only a marginally greater sensitivity (75% vs 73%), using us-qPCR as the gold standard …" ( Hofman et al JID 2019). 

"As many of the LD infections are also asymptomatic, these highly sensitive tests resurrect Koch's postulates of what defines a "pathogen", and at what concentration should a detected parasite be considered "disease-causing", necessitating the implementation of preventive or therapeutic interventions."

1) resurrect Koch's postulates - if they have not died they do not need resurrection.

2) parasitaemia is hardly physiologic - if possible the patient with parasitaemia should be treated - even if interventions do not benefit the patient directly, the patient should be treated to prevent transmission?

"It thus becomes important to assess at which point this gain in sensitivity surpasses its clinical benefit, …" Ultrasensitive diagnostics have not been developed for clinical management of patients. Please clarify this point.

The authors write at the end of the intro: "Thus these analyses aim to fill the gaps highlighted by the WHO technical committee for policy recommendations on highly sensitive point-of-care Pf malaria diagnostics by elucidating the prevalence and clinical presentation of LD parasitaemia detectable only by ultra-sensitive tools such as us-qPCR and us-mRDT and measuring its impact on clinical outcomes in febrile outpatients." Would it be possible to state clearly what the research question was?

Results:

Figure 1: please clearly indicate in the chart how many study participants were tested with 1) standard RDT, 2) us-RDT, 3) us-PCR?

Please add a table describing the demographic characteristics of 3 patient groups: low density, high density and no infections. In the table please indicate the numbers testing positive with which test (sd-RDT, us-RDT, usPCR.

If the authors insist to include the us-RDT data in the paper Table 2 is inappropriate and should be replaced with table S2.

How was "severe illness defined"?

Discussion

"LD infection prevalence fell to zero during the non-malaria season, perhaps an indication that LD infection was not functioning as an off-season reservoir, …" considering the limited sample size of the surveys the conclusions may be premature. It would take much more complete surveys of the entire population to justify the conclusion that low density infections do not contribute to transmission.

"Lower endemicity settings also have a lower diversity of parasites to which inhabitants may be more readily immune, and thus also partly explain the higher LD prevalence in recent low-endemicity settings." Shouldn't immunity clear infections and reduce the parasite prevalence?

"This study shows that, at least in a moderate malaria endemicity setting, low-density (LD) Pf parasitaemia infections undetected by mRDT did not present acute health concerns in children in an outpatient setting." Please consider rephrasing? The way the sentence currently reads the reader could think that prior to this study low density infections required acute health care. This is misleading. It would be correct to state that the study confirmed that low density infections do not require acute interventions but when detected should be treated if only to prevent transmission.

[LINK]

---

## [Decision Letter · Decision Letter 2]

14 Jul 2020

Dear Dr. Hartley,

Thank you very much for re-submitting your manuscript "Clinical relevance of low-density Plasmodium falciparum parasitaemia in untreated febrile children: a prospective cohort analysis." (PMEDICINE-D-20-00426R2) for review by PLOS Medicine.

I have discussed the paper with my colleagues and the academic editor and it was also seen again by three of the original reviewers. I am pleased to say that provided the remaining editorial and production issues are dealt with we are planning to accept the paper for publication in the journal.

[LINK]

We look forward to receiving the revised manuscript by Jul 21 2020 11:59PM. 

Sincerely,

Thomas McBride, PhD

Senior Editor 

PLOS Medicine

plosmedicine.org

Comments from the Academic Editor:

Minor points:

167 … “The arm of the trial did not affect any outcomes investigated.” Confusing, please rephrase

281 “All the children in the latter category, who were 282 put at risk of untreated HD malaria by the faulty tests were cured by day 7.” Better "spontaneously cleared the infection without treatment."

287 “As clinicians were guided by the decision support algorithms, the proportion of participants receiving antibiotics was more appropriate than what is generally observed in LMICs: This does not belong in the result section – this is part of the discussion.

“Interestingly, in our study, LD infection prevalence fell to zero during the non-malaria season, perhaps an indication that LD infection was not functioning as an off-season reservoir, albeit that larger epidemiological surveys are required before such conclusions can be made.” Indeed, this conclusion cannot be made and should not be stated in this form. Please indicate on how many samples this hypothesis is based on? How many patients were tested during the non-malaria season?

“Lower endemicity settings also have a lower diversity of parasites to which inhabitants may be more readily immune, and thus also partly explain the higher LD prevalence in recent low-endemicity settings.” Please explain what you mean by lower? Lower compared to what? do you refer to the ratio LD total number infections or the absolute prevalence. Please support this statement by appropriate references?

Since no data on HbF or HbSS are reported the discussion could be shortened by omitting the speculation related HbF and HbSS?

“diagnostic sensitivity must still be adapted to clinical relevance.” Can you please elaborate how this can be done – adapt the diagnostic sensitivity?

1- Please ensure that the study is reported according to the STROBE guideline, and include the completed STROBE checklist as Supporting Information. 1 Please add the following statement, or similar, to the Methods: "This study is reported as per the Strengthening the Reporting of Observational Studies in Epidemiology (STROBE) guideline (SChecklist)."

2- Thank you for agreeing to make your data available. At this time, please update your data availability statement to provide the URLs/accession numbers/DOIs for data access. 

3- Did your study have a prospective protocol or analysis plan? Please state this (either way) early in the Methods section.

4- Thank you for editing your title. However, please update further to: “Clinical relevance of low-density Plasmodium falciparum parasitaemia in untreated febrile children: a cohort study”

5- In the Abstract Methods and Findings, please include how the participants were recruited, the years during which the study took place, and some brief demographic information (eg, average age (SD), and sex breakdown).

6- Lines 40-43 of the Abstract could be condensed to say something to the effect of: “Treatment decisions were guided by a clinical decision support algorithm run on a mobile app, which also collected clinical data.

7- In the Abstract and Figure 1, please state how many patients were tested with standard RDT, us-RDT, and us-PCR.

8- In the Abstract and throughout, please present numerators and denominators alongside percentages.

9- Please delete the “Limitations” section header, and append the sentence on limitations to the end of the Abstract Methods and Findings section.

10- The second sentence of the Abstract Conclusion section should be worded a bit more cautiously: “These findings suggest LD parasitaemia...other infections, including those of viral origin.”

11- In the last sentence of the Abstract Conclusions, please rephrase to read: “These findings do not support a clinical benefit, nor additional risk (due to missed bacterial infections) to using ultra-sensitive malaria diagnostics at a primary care level.” 

12- Thank you for adding an Author Summary. Please reformat to follow our guidelines:

https://journals.plos.org/plosmedicine/s/revising-your-manuscript#loc-author-summary

We ask authors to provide 2-3 single sentence bullet points for each of the following questions. Bullet points should be objective, brief, succinct, specific, accurate, and avoid technical language.

Why Was This Study Done? Authors should reflect on what was known about the topic before the research was published and why the research was needed.

What Did the Researchers Do and Find? Authors should briefly describe the study design that was used and the study’s major findings. Do include the headline numbers from the study, such as the sample size and key findings. 

What Do These Findings Mean? Authors should reflect on the new knowledge generated by the research and the implications for practice, research, policy, or public health. Authors should also consider how the interpretation of the study’s findings may be affected by the study limitations.

13- In addition, please edit the last sentence of the Author Summary (currently line 88), to present your conclusions with more caution (e.g., "Our findings suggest a lack of risk or benefit in terms of clinical outcome ...").

14- Please remove trademark symbols throughout the manuscript.

15- Given that the ultra-sensitive mRDT has near identical sensitivity to standard mRDT in this cohort, the name is a bit confusing. Please mention this briefly early on (e.g., in the Introduction), and consider using quotes to acknowledge the misnomer (e.g., ‘ultra-sensitive’ mRDT).

16- The last two paragraphs of the Introduction should be rewritten to focus on clearly stating the study design and study question. As currently written details of the cohort (136-139) and study conclusions (line 146-148) should be removed, as well as the description of the “diverse and robust data”.

17- The end of the Introduction describes the data source as a randomized clinical trial, while the beginning of the Methods describes this as a “secondary analysis of a prospective cohort study”. Please clarify.

18- Please remove the “key results” icon from Table 2. Rather than color coded cells for the ND, LD, and HD cases, please use a footnote.

19- In figure 2, please describe what the shaded areas represent (95%CIs, I presume).

20- Please remove the graphical elements of Tables 3 and 4. If you wish to represent the RRs graphically, please move them to separate figures. If you choose to do so, please graphically represent the 95% CIs.

21- In the Discussion, please remove the sub-headers (e.g., “LD infection: Reservoir or Passer-by?”). Top headings (e.g., “Limitations and further research”) can remain.

22- Please rephrase the first sentence of the Discussion Conclusion: “In this study in a moderate malaria endemicity setting…”

23- At line 502, please replace “the vast majority of which being” with “likely”.

24- At line 503, please rephrase as: “Our findings suggest that implementing ultra-sensitive malaria diagnostics to detect LD infections is neither useful nor deleterious in this context.” 

25- Please remove the final sentence “To endure high quality…”

26- Please check the references carefully and make sure all use the "Vancouver" style for reference formatting, and see our website for other reference guidelines https://journals.plos.org/plosmedicine/s/submission-guidelines#loc-references

It also seems that the journal name in reference 29 is misspelt.

Comments from Reviewers:

Reviewer #1: The authors have addressed all my points.

Michael Dewey

Reviewer #3: The authors have responded satisfactorily to all my comments.

I suggest that lines 79-80 in the marked up copy be restated as follows: 

Every infection has a symptomatic threshold above which the magnitude of infection burden triggers fever and other clinical symptoms that distinguishes ill individuals from those who remain asymptomatic. 

Reviewer #4: Thank you for addressing some of my concerns. Of my 4 suggestions regarding results the authors agreed to follow one "defining severe malaria" which has also been suggested by other reviewers. None of the suggestions regarding the discussion has resulted in text changes from what I can see, although the authors kindly explain the points I missed or must have misunderstood from their perspective. I think my suggestions would have helped the reader but they do not change the findings.

[LINK]

---

## [Editor Report · Decision Letter 3]

13 Aug 2020

Dear Dr Hartley, 

On behalf of my colleagues and the academic editor, Dr. Lorenz von Seidlein, I am delighted to inform you that your manuscript entitled "Clinical relevance of low-density Plasmodium falciparum parasitaemia in untreated febrile children: a cohort study." (PMEDICINE-D-20-00426R3) has been accepted for publication in PLOS Medicine. 

PRODUCTION PROCESS

PRESS

PROFILE INFORMATION

Thank you again for submitting the manuscript to PLOS Medicine. We look forward to publishing it. 

Best wishes, 

Thomas McBride, PhD

Senior Editor 

PLOS Medicine

plosmedicine.org